# Regularized Contrastive Partial Multi-view Outlier Detection

## ABSTRACT

In recent years, multi-view outlier detection (MVOD) methods have advanced significantly, aiming to identify outliers within multi-view datasets. A key point is to better detect class outliers and class-attribute outliers, which only exist in multi-view data. However, existing methods either is not able to reduce the impact of outliers when learning view-consistent information, or struggle in cases with varying neighborhood structures. Moreover, most of them do not apply to partial multi-view data in real-world scenarios. To overcome these drawbacks, we propose a novel method named Regularized Contrastive Partial Multi-view Outlier Detection (RCPMOD). In this framework, we utilize contrastive learning to learn view-consistent information and distinguish outliers by the degree of consistency. Specifically, we propose (1) An outlier-aware contrastive loss with a potential outlier memory bank to eliminate their bias motivated by a theoretical analysis. (2) A neighbor alignment contrastive loss to capture the view-shared local structural correlation. (3) A spreading regularization loss to prevent the model from overfitting over outliers. With the Cross-view Relation Transfer technique, we could easily impute the missing view samples based on the features of neighbors. Experimental results on four benchmark datasets demonstrate that our proposed approach could outperform state-of-the-art competitors under different settings.

## CCS CONCEPTS

• **Computing methodologies** → **Anomaly detection**; • **Information systems** → *Data mining*.

## KEYWORDS

Multi-view data, Outlier detection, Unsupervised learning, Contrastive learning

## 1 INTRODUCTION

Multi-view data, which describes an entity with features sourced from various sensors or modalities, is ubiquitous in multimedia applications [12, 35, 41, 53, 58]. For example, multi-view data of a film can include textual and visual views that may capture different aspects, and multi-view data of an image can be formed by color or shape feature descriptors. Each view contributes both consensus and complementary information, enabling a more comprehensive description of the underlying data. Consequently, multi-view learning plays a crucial role in improving the generalization performance of learning models [4, 8, 20, 43, 51, 56]. However, since the quality

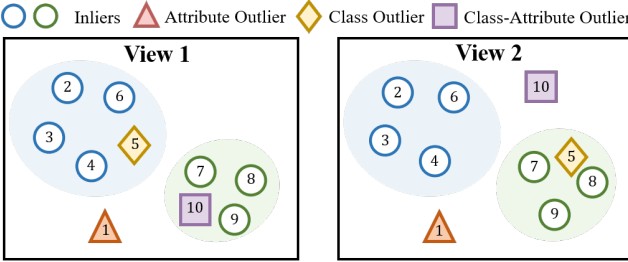

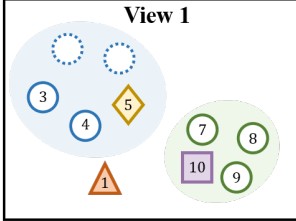 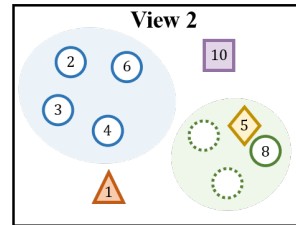

(a) Complete Multi-view Data

(b) Partial Multi-view Data

**Figure 1: Illustration of different types of outliers in complete and partial multi-view data. The dashed circles represent the missing view of an instance.**

of data collection is difficult to control, the outliers are inevitable in real-world datasets. What's worse, as the organization of multi-view data is usually more complicated, the multi-view outliers also exhibit more diverse patterns than single-view ones. Accordingly, detecting these multi-view outliers without labels becomes more challenging. As shown in Fig. 1, multi-view outliers can be sorted into three types:

- **Attribute outliers** (red triangle) are the outliers that consistently differ from most other samples in all views.
- **Class outliers** (yellow diamond) are the outliers with inconsistent features and cluster membership across different views.
- **Class-attribute outliers** (purple square) exhibit the characteristic of attribute outliers in some view while the features are inconsistent across different views.

To date, a plethora of multi-view outlier detection (MVOD) methods have been devised for this problem [2, 6, 10, 17, 24, 25, 29, 34, 47, 57]. These approaches mainly focus on the identification of multi-view-data-specific outliers, *i.e.*, class outliers and class-attribute outliers (hereinafter referred to as "class-related outliers" for brevity), given their substantial impact on overall detection efficacy. According to the ways of detecting class-related outliers, recent MVOD methods roughly fall into two categories: (1) Neighborhood similarity based methods such as NCMOD [6], SRLSP [47] and MODGD [17]. They assume that the neighborhood structures of class-related outliers are inconsistent across views, and then identify outliers by comparing the neighbors of a sample between

the view-specific and consensus similarity graphs. (2) View consistency based methods like LDSR [24] and MODDIS [19]. They assess the level of view-consistent information using latent representations, and detect class-related outliers based on the extent of view-inconsistency.

While both types of methods have demonstrated good performance, they also have their own limitations. On one hand, neighborhood similarity-based methods might struggle in scenarios where the neighborhood structures of samples exhibit significant variations. For example, when an inlier is surrounded by many class-related outliers, its neighborhood structure differs across views. On the other hand, although view consistency based methods are not affected by varying neighborhood structures, their deficiency in adequately addressing class-related outliers leads to a suboptimal performance. Since the class-related outliers exhibit a large view-inconsistency, learning from inliers and these outliers equally will hinder the model to capture the correct view-consistent and view-inconsistent information. Furthermore, the view-consistency measuring approaches in these methods often lack flexibility. For instance, in MODDIS [19], the view-consistency is simply measured by the euclidean distance between the view-specific representation and view-average representation.

Another shortcoming of existing methods is that they can only handle the complete multi-view data. Unfortunately, in real-world applications, certain views of some instances might be missing, resulting in the partial multi-view data. The missing views exacerbate the challenge of outlier detection, as the neighborhood and view consistencies are more difficult to measure, as illustrated in Fig. 1b. To effectively leverage the incomplete data, imputing the missing views becomes necessary. As an early trial, CL [13] exploits the inter-dependence across views to facilitate both view completion and outlier detection. Yet it is designed specifically for identifying class outliers. Therefore, how to better tackle the partial MVOD problem remains underexplored.

To overcome these drawbacks, we propose a novel MVOD framework, which is established on view-specific autoencoders and models the latent view consistency through contrastive learning. Considering that class-related outliers will bias the view consistency in the naïve contrastive learning, we design an outlier-aware contrastive loss with a memory bank restoring potential outliers in each mini-batch motivated by a theoretical analysis. They are then adopted as additional negative samples for contrastive learning, to push them away from inliers and mitigate their negative impact. Noticing that neighborhood structural consistency is also beneficial to promote the view consistency, we propose a neighbor alignment contrastive loss to explicitly capture the neighborhood structural consistency across views. Moreover, a spreading regularization is employed to overcome the problem of overfitting over outliers. Finally, a flexible and effective outlier scoring criteria is tailored for the proposed contrastive learning framework. With the help of neighbor alignment, we can adopt the Cross-view Relation Transfer (CRT) technique [46] for accurate missing data imputation based on the neighbor features.

In summary, our major contributions are three-fold:

- We propose a novel contrastive-learning-based partial multi-view outlier detection framework called RCPMOD, which is capable of handling partial multi-view data and simultaneously detecting three types of outliers.
- In the core of the framework, we propose an outlier-aware contrastive loss and a neighbor alignment contrastive loss to eliminate the bias caused by outliers and maximize the view consistency. We further employ a spreading regularization to overcome the problem of overfitting outliers in contrastive learning.
- With these learning techniques, we design the corresponding outlier scoring rule based on view consistency.

The effectiveness of the proposed framework is validated on four benchmark datasets under various outlier ratios and view missing rates, together with ablation and sensitivity studies.

## 2 RELATED WORK

### 2.1 Multi-view Outlier Detection

Outlier detection is an important and challenging task in machine learning [14, 48]. Currently, the majority of the methods for detecting outliers are designed for single-view data [1, 3, 21, 23, 33, 49]. In a single-view scenario, identifying outliers is relatively straightforward. These outliers are typically samples that significantly deviate from the majority, akin to attribute outliers in multi-view contexts. However, the multi-view datasets presents a more intricate situation with three types of outliers holding diverse characteristics.

In the past decade, several multi-view methods for outlier detection have been developed. Initially, the transition from single-view to multi-view outlier detection was marked by HOAD [10] which detects class outliers for the first time. Early MVOD methods [2, 10, 29] are limited by their reliance on clear cluster structure. Further advancements were made with DMOD [57], which utilizes latent coefficients and construction errors to represent multi-view data to get rid of the reliance on clear cluster structure and address both class and attribute outliers simultaneously. Following DMOD, MLRA [25], MLRA+ [26] and MuvAD [34] are proposed, improving the performance of outlier detection but they are only capable of handling data with two views.

To overcome the limitations on the view number, LDSR [24] divides representations into view-consistent and view-inconsistent parts and quantifies the degree of inconsistency by the value of the view-inconsistent parts to detect outliers. Additionally, it first raises the concept of class-attribute outliers. Adopting a similar paradigm as LDSR [24], MODDIS [19] focuses on dividing representations in a deep learning way by using separate networks to learn view-consistent and view-inconsistent parts, respectively.

Recently, newer methods based on neighborhood similarity were developed. NCMOD [6], leveraging an autoencoder network, maps samples to a latent space for each view and constructs neighborhood consensus graphs to detect outliers. SRLSP [47] also constructs neighbor similarity graphs and fuses them with a graph fusion term. MODGD [17] then pays attention to outliers when fusing neighborhood graphs of views through introducing a row-wise sparse outlier matrix characterizing outliers in data.

**Partial multi-view outlier detection.** The MVOD problem is underexplored when some views of data are missing. To the best of our knowledge, there is only one early trial, *i.e.,* CL [13], tailored for this task. It proposes a Collective Learning based framework

that exploits inter-dependence among different views for view completion and outlier detection. However, CL could only handle class outliers and fails when facing attribute outliers.

## 2.2 Contrastive Multi-view Learning

Contrastive learning stands out as a notable method in unsupervised representation learning [30, 37, 40, 44]. It learns intrinsic information of unsupervised data by enhancing the similarity between positive pairs and reducing it among negative pairs. This approach has been successfully applied in various fields, including computer vision [5, 7, 16, 27], natural language processing [11, 22] and audio processing [31]. The method has also been extended to multi-view learning, with significant works in this area including [15, 38]. A representavtive work is [38], which introduces a multi-view coding framework using contrastive learning to understand scene semantics better. Recent efforts have been made to explore the implementations of contrastive learning in multi-view clustering [28, 39, 45, 52, 54]. For example, MFLVC [52] combines instance- and cluster-level contrastive learning on high-level features to learn more common semantics across views, AGCL [45] adopt within-view graph contrastive learning and cross-view graph consistency learning to learn more discriminative representations for clustering.

In this paper, we utilize contrastive learning in MVOD to pursue the cross-view consistency, with some special designs to alleviate the influence of outliers. Meanwhile, a neighbor alignment contrastive module is designed to further learn the neighborhood structural consistency and improve the imputation performance on partial multi-view datasets.

## 3 METHODOLOGY

### 3.1 Problem Setting

Without loss of generality, we take bi-view data as an example. Consider a partial bi-view dataset $X_{ms} = \{X_c^{(1)}, X_c^{(2)}, X_a^{(1)}, X_b^{(2)}\}$ without labels, where $\{X_c^{(1)}, X_c^{(2)}\}$ denote the instances presented in both views (also called complete data subset) with the size of $N$, $X_a^{(1)}$ and $X_b^{(2)}$ denote those presented in one view but missing in the other view. Let $X^{(1)} = \{X_c^{(1)}, X_a^{(1)}\}$ and $X^{(2)} = \{X_c^{(2)}, X_b^{(2)}\}$ be all the samples in view 1 and 2 with a size of $N_1$ and $N_2$, respectively. The data might simultaneously contain attribute/class/class-attribute outliers. Our target is designing a scoring function $s(\cdot)$ to detect outliers in the data in an unsupervised manner, with a higher score indicating a larger probability to be abnormal.

### 3.2 Outlier-aware Contrastive Learning

Following the convention of deep unsupervised multi-view learning [28, 52], we adopt the autoencoder (AE) to learn the latent representation of each views. Let $f^{(v)}$ and $g^{(v)}$ denote the encoder and decoder for the $v$-th view, respectively. To preserve the information of each view in the latent space, the AE reconstruction loss is defined as:

$$\mathcal{L}_{ar} = \frac{1}{2} \sum_{v=1}^{2} \sum_{i=1}^{N_v} \left\| x_i^{(v)} - g^{(v)}\left(f^{(v)}\left(x_i^{(v)}\right)\right) \right\|_2^2, \tag{1}$$

where $x_i^{(v)}$ denotes the $i$-th sample in $X^{(v)}$. Hence, the latent representation of $x_i^{(v)}$ is given by $z_i^{(v)} = f^{(v)}(x_i^{(v)})$.

To facilitate the multi-view outlier detection, we hope to learn a latent space in which inliers exhibit a large cross-view consistency while outliers (especially class-related ones) are quite the opposite. In many recent multi-view learning methods [39, 52], the view-consistent information can be learned by contrastive learning. It pulls the embeddings of the same instance in each view close to each other while simultaneously pushing away those of different instances. For a given latent representation $z_i^{(1)}$, its counterpart in the other view $z_i^{(2)}$ is considered as the positive sample, and the rest samples in all views usually serve as negative samples. Using the cosine similarity $s(x, y)$, a typical multi-view contrastive loss could be formulated as:

$$\mathcal{L}_{con} = -\frac{1}{2} \sum_{m=1}^{2} \sum_{i=1}^{N} \log \frac{e^{s(z_i^{(m)}, z_i^{(m')})/\tau_F}}{\sum_{j=1}^{N} \sum_{v=1}^{2} e^{s(z_i^{(m)}, z_j^{(v)})/\tau_F}}, \tag{2}$$

where $m'$ is the counterpart view of $m$ (e.g., $m' = 2$ when $m = 1$), and $\tau_F$ denotes the temperature parameter.

However, the naïve contrastive loss overlooks the presence of outliers. Given that class-related outliers usually exhibit a large inconsistency among different views, arbitrarily pursuing the view-consistency for all the contaminated data will inevitably bias the latent space and then harm the learning. Recall that the contrastive loss fundamentally maximizes a lower bound on the mutual information between different views of an instance [40], i.e., $I(z^{(1)}, z^{(2)})$. But in our case, we should only maximize the mutual information for inliers and keep the mutual information of outliers low to alleviate their negative impact. According to the characteristic of class-related outliers, we can naturally assume that the mutual information between different views of class-related outliers is upper-bounded:

$$I(x_o^{(1)}, x_o^{(2)}) \leq \varepsilon, \tag{3}$$

where $x_o^{(1)}$ and $x_o^{(2)}$ represent the different views of any arbitrary class-related outlier. Then we can find that a lower bound exists for the contrastive loss of such outliers, as shown in the following proposition. Due to space limitations, we leave the detailed proof in the supplementary materials.

PROPOSITION 3.1. *If $I(x_o^{(1)}, x_o^{(2)}) \leq \varepsilon$, then the contrastive loss value of outlier instances is lower-bounded by $\log(2N) - \varepsilon$.*

PROOF SKETCH. Following [40], it is easy to show that:

$$I(z_o^{(1)}, z_o^{(2)}) \geq \log(2N) - \mathcal{L}_{con}^o, \tag{4}$$

where $\mathcal{L}_{con}^o$ denotes the contrastive loss over all the outliers but the negative samples could be chosen from both inliers and outliers.

Meanwhile, by the data processing inequality, we have:

$$I(z_o^{(1)}, z_o^{(2)}) \leq I(x_o^{(1)}, x_o^{(2)}) \leq \varepsilon. \tag{5}$$

Combining the above results, we can obtain:

$$\mathcal{L}_{con}^o \geq \log(2N) - I(z_o^{(1)}, z_o^{(2)}) \geq \log(2N) - \varepsilon. \tag{6}$$

$\square$

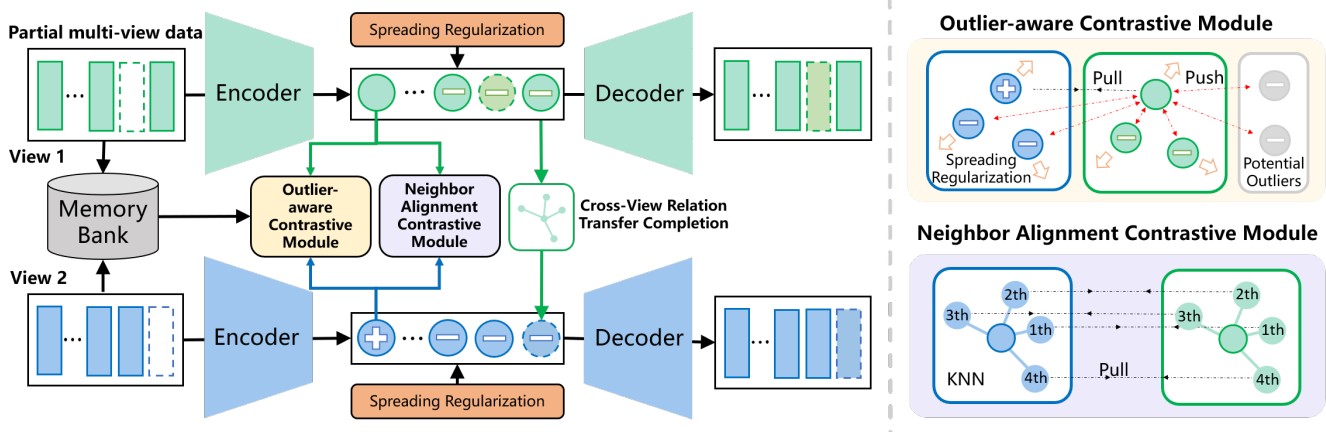

**Figure 2: Overview of RCPMOD on bi-view data. Two key contrastive learning modules are applied on the latent space to promote the view consistency: (1) In outlier-aware contrastive module, potential class-related outliers are restored in a memory bank and used as additional negative samples. (2) In neighbor alignment contrastive module, the corresponding neighbors of a sample are aligned to learn the cross-view structural correlations. Moreover, we adopt a spreading regularization to prevent from overfitting on class-related outliers. The missing samples are imputed by the Cross-view Relation Transfer technique.**

The lower bound given in Proposition 3.1 suggests the feasibility of identifying outliers based on their loss values. Indeed, the contrastive loss value of each instance could also reflect how it is consistent across different views during the learning. Class-related outliers, being predominantly view-inconsistent, may exhibit higher loss values compared to inliers. In this sense, it is also natural to adopt this value as the indicator of such outliers. For computational convenience, here we simplify the calculation in Eq.(2), and only adopt the cross-view cosine similarity of each view-complete instance, i.e., $s(z_i^{(1)}, z_i^{(2)})$, as the criterion. To utilize these potential outliers, we propose employing a memory bank to store them. These potential outliers could be used as negative samples for each $z_i^{(v)}$. In practice, we select a fixed ratio $\eta$ of instances with the smallest cross-view similarities in each mini-batch to form the memory bank $\mathcal{M}$ with a size of $N_M$. The memory bank is a first-in-first-out queue to keep the potential outliers up-to-date. By incorporating the newly formed negative pairs into Eq.(2), we formulate the outlier-aware contrastive loss as:

$$\mathcal{L}_{oa} = -\frac{1}{2} \sum_{m=1}^{2} \sum_{i=1}^{N} \log \frac{e^{s(z_i^{(m)}, z_i^{(m')})/\tau_F}}{\sum_{j=1}^{N} \sum_{v=1}^{2} e^{s(z_i^{(m)}, z_j^{(v)})/\tau_F} + P_M},$$

$$P_M = \sum_{v=1}^{2} \sum_{t=1}^{N_M} e^{s(z_i^{(m)}, m_t^{(v)})/\tau_F},$$ (7)

where $m_t^{(v)}$ is the $t$-th sample representation in $v$-th view in $\mathcal{M}$. With this modified contrastive loss, class-related outliers are more distinguishable in view consistency.

Note that to accurately learn the latent space, the outlier-aware contrastive learning is only conducted on the view-complete instances at the beginning of training. After training for few epochs, we start to impute the missing view samples (the details will be introduced later) and then apply Eq.(7) to both the complete subset and imputed data.

## 3.3 Neighbor Alignment Contrastive Learning

It is often assumed that data in different views share abundant local structural correlation. This information is apparently helpful in identifying class-related outliers since them usually exhibit inconsistent local structure across views. However, the standard contrastive learning objective is not able to exploit such information. To address this, we design a contrastive loss to explicitly learn the cross-view local neighborhood correlation by aligning the representations of $K$-nearest neighbors of an instance in different views. Specifically, for each sample $z_i^{(v)}$, we find its $K$-nearest neighbors ($K$-NNs) $\{z_{i,t}^{(v)}\}_{t=1}^{K}$ within the same view, where $z_{i,t}^{(v)}$ denote the $t$-th neighbor of $z_i^{(v)}$. The neighbor alignment contrastive loss could then be formulated as:

$$\mathcal{L}_{na}^{t} = -\frac{1}{2} \sum_{m=1}^{2} \sum_{i=1}^{N} \log \frac{e^{s(z_{i,t}^{(m)}, z_{i,t}^{(m')})}}{\sum_{j=1}^{N} \sum_{v=1}^{2} e^{s(z_{i,t}^{(m)}, z_{j,t}^{(v)})}},$$

$$\mathcal{L}_{na} = \frac{1}{K} \sum_{t=1}^{K} \mathcal{L}_{na}^{t}.$$ (8)

It is noteworthy that since the $K$-nearest neighbors are calculated within individual views, the neighbor sets $\{z_{i,t}^{(1)}\}_{t=1}^{K}$ and $\{z_{i,t}^{(2)}\}_{t=1}^{K}$ are not necessarily identical. As shown in the right panel of Fig. 2, the proposed loss encourages the corresponding nearest neighbors across different views of an instance to be close. By doing so over all $K$ nearest neighbors, the neighborhood structure of each instance is aligned across different views, which further enhances the view-consistency.

Besides, in the beginning of training, the network usually cannot capture a stable latent structure in the data. Thus, the $K$-NNs in this stage are obtained based on the input features. When the latent structure becomes stable, the neighbors are then updated based on the newest latent features.

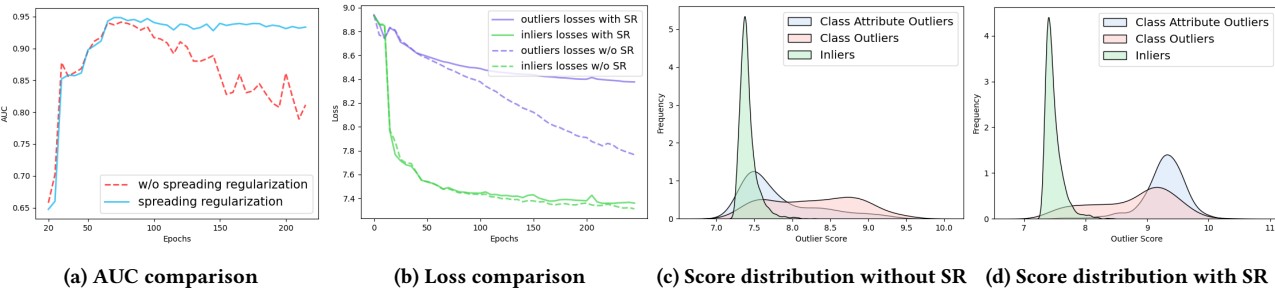

(a) AUC comparison     (b) Loss comparison     (c) Score distribution without SR    (d) Score distribution with SR

Figure 3: (a) Comparison of the detection AUC with and without spreading regularization (SR) on SCENE15. (b) Comparison of the average loss value over inliers and outliers. (c)/(d) Outlier score distribution without/with SR.

## 3.4 Spreading Regularization

The above two contrastive losses equip our model with a strong ability to learn the view-consistent information in the presence of outliers, which is helpful for the detection. However, learning with contrastive losses may also incur some side effects. As the dotted red lines in Fig. 3a show, although the detection performance increases rapidly at the beginning of training, it then tends to decrease after reaching the performance peak. Such an overfitting could be further demonstrated through the dashed lines in Fig. 3b. Apparently, the cross-view consistency is much easier to achieve over inliers than outliers, so the contrastive loss of inliers decreases much faster. Unfortunately, as the learning goes on, the inliers are sufficiently view-consistent, turning the model's attention to promote the consistency over outliers. Accordingly, the loss of class-related outliers starts to decrease rapidly when the loss of inliers gradually becomes stable. On the other hand, due to the underlying clustering effect of contrastive losses [18], outliers might become still closer and closer to inliers in the latent space. This intrinsic trend cannot be completely alleviated by the outlier-aware design in Sec. 3.2 due to the limited volume of the outlier memory bank. It will also result in the outliers, especially attribute-related outliers, becoming increasingly indistinguishable.

To overcome this issue, we need to control the closeness for samples. We extend the KoLeo loss [32] into the multi-view setting as a regularizer of contrastive losses:

$$\mathcal{L}_{\text{KoLeo}} = -\frac{1}{2} \sum_{v=1}^{2} \sum_{i=1}^{N_v} \log(\delta_i^{(v)}), \tag{9}$$

where

$$\delta_i^{(v)} = \min_{j \neq i} \|z_i^{(v)} - z_j^{(v)}\|. \tag{10}$$

Here the closest points in each view are pushed away, which continuously scatters the latent representations. Following [32], a rank preserving loss is also adopted to prevent the KoLeo loss from undermining the latent structure:

$$\mathcal{L}_{\text{rank}} = -\frac{1}{2} \sum_{v=1}^{2} \sum_{i=1}^{N_v} \max\left(0, \|z_i^{(v)} - z_i^{(v)+}\|_2 - \|z_i^{(v)} - z_i^{(v)-}\|_2\right), \tag{11}$$

where the positive sample $z_i^{(v)+}$ is randomly chosen among the $k_{pos}$ nearest neighbors of $z_i^{(v)}$ and the negative sample $z_i^{(v)-}$ is the $k_{neg}$-th neighbor. $k_{neg}$ is usually set as a much larger value than $k_{pos}$ so that $z_i^{(v)+}$ and $z_i^{(v)-}$ can be near and far from $z_i^{(v)}$, respectively. This loss mainly focuses on preserving the neighborhood structure in each view, so that the KoLeo loss will not break the data structure.

Thus the spreading regularization loss can be formulated as:

$$\mathcal{L}_{sr} = \mathcal{L}_{\text{KoLeo}} + \mathcal{L}_{\text{rank}}. \tag{12}$$

With the help of this regularization, the detection performance could be significantly stabilized and the overfitting on outliers are prevented, as shown by the solid lines in Fig. 3a and 3b. Furthermore, the outlier score distribution before and after adding spreading regularization in Fig. 3c and 3d also demonstrates the effect of this loss. We can find that the overlapping between inliers and outliers is reduced with spreading regularization.

Putting all together, the overall learning objective of RCPMOD can be formulated as:

$$\mathcal{L} = \mathcal{L}_{ar} + \lambda_1 \mathcal{L}_{oa} + \lambda_2 \mathcal{L}_{na} + \mu \mathcal{L}_{sr}, \tag{13}$$

where $\lambda_1$, $\lambda_2$, $\mu$ are balancing parameters. This framework could be easily extend to the case with more than two views similar to existing multi-view learning method such as [52].

## 3.5 Outlier Scoring

The design for a proper outlier scoring function should consider the characteristics of the three kinds of outliers. In our framework, we mainly have the following consideration:

- For attribute outliers, as they are abnormal in all views and dissimilar with the majority, they are usually harder for AEs to reconstruct than inliers. Hence, a large reconstruction error can indicate an attribute outlier.
- For class-outliers, as analyzed in Sec. 3.2, they are largely view-inconsistent. With the optimization of outlier-aware contrastive loss, the view-consistency of normal instances will gradually increase while that of class outliers remain at a very low level. Such a gap will be reflected in the value of the contrastive loss and a large contrastive loss value can indicate a class-outlier.
- For class-attribute outliers, as it contains the characteristics of both attribute and class outliers, the combination of the

**Table 1: Data statistics of the benchmark datasets.**

| Datasets | Instances | Views | Classes |
|----------|-----------|-------|---------|
| BDGP | 2500 | 2 | 5 |
| SCENE15 | 4568 | 3 | 15 |
| LandUse21 | 2100 | 3 | 21 |
| Fashion | 10000 | 3 | 10 |

**Table 2: Different combinations of outlier ratios.**

| id | $\rho_1$ | $\rho_2$ | $\rho_3$ |
|----|----------|----------|----------|
| 1 | 0.02 | 0.05 | 0.08 |
| 2 | 0.02 | 0.08 | 0.05 |
| 3 | 0.05 | 0.02 | 0.08 |
| 4 | 0.05 | 0.08 | 0.02 |
| 5 | 0.08 | 0.02 | 0.05 |
| 6 | 0.08 | 0.05 | 0.02 |

reconstruction error and contrastive loss value is able to indicate outliers in this type.

Then we could obtain the corresponding scoring function as:

$$s(x_i) = s_r(x_i) + s_c(x_i), \tag{14}$$

where

$$s_r(x_i) = \frac{1}{2} \sum_{v=1}^{2} \left\| x_i^{(v)} - \hat{x}_i^{(v)} \right\|_2^2,$$

$$s_c(x_i) = -\frac{1}{2} \sum_{m=1}^{2} \log \frac{e^{d(z_i^{(m)}, z_i^{(m')})/\tau_F}}{\sum_{j=1}^{N} \sum_{v=1}^{2} e^{d(z_i^{(m)}, z_j^{(v)})/\tau_F}}. \tag{15}$$

Here $s_r(x_i)$ is the reconstruction error across all the views, which will be large for attribute outliers; $s_c(x_i)$ is the contrastive loss value and should be large for class outliers. For partial data, $s_c(x_i)$ is calculated after imputation. Meanwhile, class-attribute outliers will also have large $s(x_i)$s. What's more, the inliers are easy to reconstruct and their view-consistency should be good, resulting in a small $s(x_i)$.

**Missing Sample Imputation.** With the aligned neighborhood structure, our method can easily recover the representation of missing samples with the Cross-view Relation Transfer technique [46]. The core idea is to impute the missing view based on the nearest neighbors in other views. Taking the recovery of $z_{b,i}^{(1)}$ as an example. We first obtain the $K$ nearest neighbors of $z_{b,i}^{(2)}$ in view 2 and find their counterparts in view 1. Since some neighbor counterparts may be missing in view 1, we ignore these missing samples and take the average of the rest complete ones as the recovered latent representation $\hat{z}_{b,i}^{(1)}$.

## 4 EXPERIMENTS

### 4.1 Experimental Settings

**Datasets and evaluation protocols.** Four widely-used datasets were used in our experiments. Among the selected datasets, BDGP

[42] is a drosophila embryos dataset, with each of the instances represented by visual and textual features as two views. SCENE15 [9] consists of images of natural scenes where each image is described by GIST, SIFT, and LBP features. LandUse21 [55] contains satellite images from with PHOG and LBP features. Fashion [50] is a novel image dataset of fashion products which treat different three styles as three views of one product. The details are recorded in Table 1. For a simpler notation, we denote LandUse-21, Scene15, BDGP and Fashion as 'L', 'S', 'B' and 'F' respectively for short.

Following the previous work [17, 19, 24, 47], we generate outliers in these datasets with the following strategy: (1) For attribute outliers, we randomly choose an instance, and replace its feature in all views by random values. (2) For class outliers, we randomly take some pairs of instances and swap the feature vectors in $\lfloor \frac{V}{2} \rfloor$ views while keeping feature vectors in the other views unchanged. (3) For class-attribute outliers, we also randomly choose some pairs of instances, swap feature vectors in $\lfloor \frac{V}{2} \rfloor$ views, and replacing features with random values in the other views. Also, we vary the outlier ratio for a more comprehensive evaluation. Table 2 illustrates the different combinations for ratios of attribute outlier ($\rho_1$), class outlier ($\rho_2$) and class-attribute outlier ($\rho_3$).

Besides, as the original datasets are all complete, we follow [13] to form partial multi-view data by randomly removing one view of some randomly selected instances. The view missing rate is defined as $\frac{N_{all} - N}{N_{all}}$, where $N_{all}$ is the total number of instances involved in partial multi-view data. To evaluate the ability of dealing different degree of view missing, we evaluate the methods on the missing rate of 0, 0.15, 0.3, 0.45, respectively. It is noteworthy that we also use complete multi-view datasets for evaluation, to show the strength of the proposed method in an ideal case.

**Baselines.** We compare our method with five multi-view outlier detection methods including MODDIS [19], NCMOD [6], SRLSP [47], MODGD [17] and CL [13]. Among them, the first four models are merely designed for complete multi-view data. So for these methods, partial multi-view data is imputed using the method proposed by a recent incomplete multi-view learning framework DSIMVC [36] for a fair comparison.

**Implementation details.** The structures of autoencoders are slightly different for the datasets. For LandUse21 and Scene15, we use three fully-connected layers as the encoder, and their latent dimensions are 1024-1024-64. For BDGP and Fashion, the depth of the encoder is 2, and the structure is 1024-64 and 1024-256, respectively. The decoders then have a reverse structure. The activation function is ReLU. The Adam optimizer is adopted with the learning rate of $1e^{-3}$ for training. The hyperparameter $\lambda_1$ and $\lambda_2$ are fixed to 1 and $\eta$ is fixed to 0.05. The number of nearest neighbors $K$ is set to 6 for all datasets. We design a piecewise-linear scheduler for $\mu$ to adjust the impact of SR. In the first 100 epochs, $\mu$ increases from 0 to a specific value $\mu_1$ linearly, and then rises to a larger value $\mu_2$ in the rest epochs. $\mu_1/\mu_2$ is set as 0.01/0.2, 0.02/0.2, 0.02/0.4, 0.05/0.4 on BDGP, LandUse21, Scene15 and Fashion, respectively.

### 4.2 Comparisons with Baseline Methods

The detection AUC results under different missing rates are recorded in Table 3 and 4. The dataset name is shorted and combined with

**Table 3: The detection AUC (%) on different datasets under the missing rates of 0 and 0.15. The value marked in "red" holds the highest value, and "blue" holds the second highest.**

**(a) AUC on BDGP and SCENE15 with no missing view**

|    | CL | MODDIS | NCMOD | SRLSP | MODGD | Ours |
|----|----|--------|-------|-------|-------|------|
| B1 | 49.84±1.53 | 88.64±0.92 | 86.03±1.22 | 91.29±1.22 | 76.69±1.56 | **97.05±0.18** |
| B2 | 52.15±1.23 | 80.85±1.23 | 77.18±1.10 | 85.14±0.91 | 69.62±1.62 | **95.67±0.65** |
| B3 | 47.28±1.80 | 95.58±0.51 | 94.05±0.78 | **96.62±0.44** | 86.13±1.86 | 95.80±0.72 |
| B4 | 51.33±0.79 | 81.45±1.31 | 78.29±0.74 | 85.38±0.78 | 71.53±1.41 | **91.30±0.48** |
| B5 | 50.17±2.49 | 95.83±0.45 | 94.01±1.15 | **96.66±0.42** | 88.52±0.74 | 95.58±0.54 |
| B6 | 51.33±3.22 | 88.27±0.71 | 86.80±1.83 | 91.29±1.28 | 82.09±1.14 | **92.18±1.00** |
| S1 | 52.25±4.89 | 92.24±0.40 | 91.12±1.09 | 95.89±0.21 | 85.30±1.16 | **97.67±0.41** |
| S2 | 54.73±4.18 | 87.40±0.67 | 82.78±1.20 | 93.32±0.49 | 76.29±0.78 | **95.03±0.46** |
| S3 | 53.33±3.41 | 95.50±0.40 | 95.08±0.38 | 92.98±0.37 | 93.83±0.45 | **97.89±0.57** |
| S4 | 53.55±3.89 | 87.27±0.88 | 83.61±2.88 | 93.20±0.45 | 76.39±0.98 | **94.61±0.69** |
| S5 | 51.47±3.11 | 94.54±2.35 | 95.98±0.53 | 93.80±0.33 | 93.68±0.32 | **97.36±0.31** |
| S6 | 52.20±2.85 | 92.03±0.60 | 89.44±1.32 | 95.85±0.27 | 85.19±1.28 | **97.02±0.52** |

**(b) AUC on BDGP and SCENE15 with a missing rate of 0.15**

|    | CL | MODDIS | NCMOD | SRLSP | MODGD | Ours |
|----|----|--------|-------|-------|-------|------|
| B1 | 50.37±1.54 | 87.97±1.01 | 86.00±0.76 | 88.58±1.25 | 75.11±1.82 | **97.09±0.27** |
| B2 | 49.11±2.01 | 80.77±0.33 | 78.16±0.65 | 82.71±1.22 | 69.47±1.81 | **95.27±0.74** |
| B3 | 50.21±1.81 | 95.31±0.33 | 93.80±0.34 | 95.22±0.81 | 83.76±1.04 | **96.79±0.59** |
| B4 | 49.86±2.35 | 81.33±1.28 | 79.37±0.66 | 83.74±0.59 | 72.10±1.17 | **89.34±2.21** |
| B5 | 47.24±5.33 | 95.32±0.29 | 94.42±0.41 | 95.75±0.46 | 88.35±0.80 | **95.90±0.31** |
| B6 | 47.02±4.59 | 88.26±0.57 | 88.41±0.55 | 89.77±0.64 | 82.24±0.48 | **91.80±1.09** |
| S1 | 48.95±3.66 | 92.10±0.99 | 87.66±0.72 | 95.22±0.69 | 83.40±0.59 | **96.31±0.23** |
| S2 | 49.81±4.41 | 86.94±0.41 | 82.04±2.09 | 92.38±0.37 | 74.07±1.30 | **96.39±0.47** |
| S3 | 48.84±3.19 | 96.08±0.36 | 94.66±0.61 | 93.75±0.36 | 93.26±0.42 | **97.08±0.30** |
| S4 | 48.55±3.70 | 87.40±0.91 | 81.29±0.84 | 92.68±0.57 | 74.66±1.14 | **93.95±1.44** |
| S5 | 50.16±2.12 | 95.81±0.23 | 95.02±0.17 | 94.26±0.27 | 93.54±0.40 | **96.37±0.12** |
| S6 | 49.76±2.38 | 92.57±0.85 | 88.86±1.40 | 95.75±0.84 | 84.02±0.39 | **96.40±0.43** |

**(c) AUC on Fashion and LandUse21 with no missing view**

|    | CL | MODDIS | NCMOD | SRLSP | MODGD | Ours |
|----|----|--------|-------|-------|-------|------|
| F1 | 47.35±3.30 | 91.68±0.46 | 90.68±0.39 | 93.22±0.40 | 84.09±0.41 | **97.63±0.09** |
| F2 | 48.19±2.87 | 86.04±0.51 | 86.39±0.41 | 88.52±0.52 | 74.23±0.33 | **96.55±0.36** |
| F3 | 47.78±6.16 | 96.44±0.20 | 96.20±0.35 | 97.52±0.16 | 93.54±0.19 | **98.61±0.17** |
| F4 | 48.00±3.81 | 86.57±0.37 | 86.92±0.57 | 88.59±0.58 | 74.35±0.49 | **96.09±0.42** |
| F5 | 45.87±8.04 | 96.75±0.12 | 96.70±0.18 | 97.52±0.18 | 93.57±0.14 | **98.34±0.19** |
| F6 | 47.03±5.62 | 92.07±0.45 | 92.09±0.48 | 93.29±0.43 | 84.15±0.40 | **96.67±0.32** |
| L1 | 54.50±10.52 | 91.34±0.43 | 86.77±0.76 | 93.88±0.71 | 89.15±0.38 | **98.02±0.36** |
| L2 | 53.97±10.14 | 85.41±1.06 | 78.18±0.98 | 89.89±0.58 | 82.38±1.32 | **97.76±0.50** |
| L3 | 53.34±9.93 | 96.52±0.47 | 94.52±0.73 | 97.82±0.44 | 95.66±0.56 | **98.94±0.21** |
| L4 | 53.39±8.59 | 85.61±0.79 | 78.40±0.86 | 89.81±0.60 | 82.18±1.33 | **97.36±0.24** |
| L5 | 53.77±7.92 | 96.56±0.51 | 95.09±1.57 | 97.85±0.46 | 95.63±0.44 | **99.06±0.29** |
| L6 | 52.95±9.40 | 91.16±0.58 | 85.65±0.46 | 93.88±0.76 | 89.23±0.65 | **97.61±0.79** |

**(d) AUC on Fashion and LandUse21 with a missing rate of 0.15**

|    | CL | MODDIS | NCMOD | SRLSP | MODGD | Ours |
|----|----|--------|-------|-------|-------|------|
| F1 | 46.37±5.68 | 90.93±0.35 | 91.62±0.24 | 92.32±0.17 | 83.58±0.18 | **97.70±0.07** |
| F2 | 47.62±4.05 | 86.76±0.86 | 87.05±0.34 | 88.31±0.44 | 75.07±1.40 | **96.66±0.28** |
| F3 | 45.30±8.86 | 96.14±0.38 | 94.79±0.35 | 96.85±0.39 | 92.01±1.20 | **98.55±0.13** |
| F4 | 46.83±4.90 | 87.38±0.31 | 87.90±0.33 | 88.55±0.57 | 74.68±0.44 | **96.04±0.17** |
| F5 | 44.59±10.48 | 96.23±0.52 | 96.08±0.42 | 96.95±0.32 | 92.62±1.87 | **98.29±0.14** |
| F6 | 45.07±9.30 | 92.39±0.34 | 92.09±0.78 | 93.03±0.21 | 82.62±1.92 | **97.01±0.46** |
| L1 | 50.82±9.81 | 90.72±0.62 | 85.47±0.35 | 93.04±0.79 | 87.39±0.54 | **97.05±0.35** |
| L2 | 50.23±9.28 | 86.05±0.97 | 77.70±0.88 | 89.43±0.98 | 79.90±1.22 | **96.78±0.39** |
| L3 | 50.62±9.00 | 96.16±0.19 | 93.78±0.30 | 97.27±0.39 | 94.84±0.34 | **97.37±0.60** |
| L4 | 51.25±9.38 | 86.25±1.16 | 77.77±1.14 | 89.92±1.14 | 80.51±1.36 | **95.67±1.36** |
| L5 | 51.92±9.51 | 96.26±0.21 | 94.35±0.50 | 97.25±0.39 | 94.91±0.49 | **98.32±0.35** |
| L6 | 50.45±7.88 | 91.10±0.87 | 85.80±0.43 | 93.50±1.05 | 88.15±0.91 | **97.03±0.32** |

the setting id denoted in Table 2. From these tables we have the following observations:

- RCPMOD outperforms all the baseline methods in most settings, regardless of whether the dataset is partial or not. Among all datasets, our method achieves best performance on Fashion, surpassing the second best models in all settings with a relative improvement of up to 9.1%.
- When there are more class outliers (*i.e.,* setting 2 and 4), the performance of competitors is obviously degenerated. This is mainly due to their lacking of attention to class outliers or the inability of detecting class outliers in boundary situations. In contrast, our method could achieve much higher AUCs on these settings, which indicates the superiority of our method when detecting class outliers. The performance degradation of baselines under different ratios of class-attribute outliers is less obvious. The reason might be that such outliers are also detectable based on their abnormal attributes in some views. Nevertheless, our method still outperforms the baselines in settings with more class-attribute outliers (*i.e.,* setting 1 and 3), which can be attributed to the enhanced detection of class-attribute outliers based on view inconsistency.
- Despite CL can directly deal with partial multi-view data, it is originally designed only for the detection of class outliers. This results in its poor performance in the presence of attribute and class-attribute outliers.

## 4.3 Sensitivity Analysis

Our method contain several important hyperparameters including the balancing factor $\lambda_1$, $\lambda_2$, $\mu$, and the sampling rate $\eta$ for the memory bank. To analyze the hyperparameter sensitivity of RCPMOD, we fix the missing rate to 0.3 and the outlier ratio of all types of outliers to 0.05 and evaluate RCPMOD using different values of $\lambda_1$, $\lambda_2$, $\mu$ and $\eta$. As a scheduler of $\mu$ is adopted in the training , we only vary $\mu_1$ used in the warm-up stage which empirically has more impact on the results.

**Impact of $\lambda_1$ and $\lambda_2$.** As shown in the first two subplots of Fig. 4, a relatively large value of $\lambda_1$ and $\lambda_2$ would be beneficial. But when they are assigned with excessively large values with $\mu$ unchanged, the performance of RCPMOD will significantly decrease due to the overfitting to outliers.

**Impact of $\eta$.** From the third subplot, we see that the performance is relatively stable within the whole range. Note that the curves roughly peak at an $\eta$ value of 0.05 or 0.1, which is close to the ratio of class-related outliers in datasets.

**Impact of $\mu$.** The last subplot of Fig. 4 demonstrates the performance tends to decrease when this value is increased. Apparently it shows that a large $\mu$ is not a good choice, suggesting that arbitrarily pushing away the points can negatively affect both the performance and stability of the model.

## 4.4 Ablation Study

The ablation results of each loss module are shown in Table 5. From ablated variants (C), (D) and (E), we can observe that removing

**Table 4: The detection AUC (%) on different datasets under the missing rates of 0.3 and 0.45.**

**(a) AUC on BDGP and SCENE15 with a missing rate of 0.3**

| | CL | MODDIS | NCMOD | SRLSP | MODGD | Ours |
|---|---|---|---|---|---|---|
| B1 | 50.31±3.06 | 88.02±0.66 | 85.90±0.58 | 87.22±0.62 | 72.20±1.57 | **96.97±0.46** |
| B2 | 50.72±3.45 | 81.20±0.89 | 79.25±1.17 | 82.09±0.81 | 66.15±1.23 | **95.17±0.83** |
| B3 | 49.34±1.86 | 95.35±0.78 | 94.75±0.66 | 93.39±1.68 | 80.95±1.90 | **96.83±0.32** |
| B4 | 49.92±1.90 | 81.73±0.62 | 80.75±0.19 | 83.93±1.23 | 70.01±0.97 | **89.48±3.08** |
| B5 | 48.69±3.53 | 95.62±0.23 | 94.18±0.79 | 94.79±0.62 | 86.07±1.03 | **96.69±0.55** |
| B6 | 48.03±2.85 | 88.32±0.43 | 87.46±0.52 | 89.57±0.63 | 80.87±1.49 | **92.30±1.14** |
| S1 | 47.68±2.70 | 91.39±0.54 | 87.88±1.29 | 94.07±0.69 | 81.66±0.41 | **96.06±0.61** |
| S2 | 48.01±2.33 | 86.90±1.24 | 81.36±1.36 | 90.67±1.74 | 74.14±4.52 | **96.10±0.27** |
| S3 | 46.81±2.47 | 94.59±0.88 | 95.69±0.82 | 93.98±0.38 | 92.21±0.16 | **96.21±0.77** |
| S4 | 48.07±2.69 | 87.65±1.42 | 81.59±1.12 | 91.33±1.98 | 74.73±2.44 | **94.40±0.66** |
| S5 | 47.97±1.83 | 94.39±1.42 | 95.29±0.43 | 94.51±0.43 | 93.18±0.20 | **96.74±0.43** |
| S6 | 48.50±1.25 | 92.51±0.72 | 89.85±0.57 | 94.58±1.25 | 83.48±0.38 | **95.69±0.33** |

**(b) AUC on BDGP and SCENE15 with a missing rate of 0.45**

| | CL | MODDIS | NCMOD | SRLSP | MODGD | Ours |
|---|---|---|---|---|---|---|
| B1 | 50.28±3.62 | 87.24±0.72 | 86.31±0.56 | 85.81±0.79 | 69.15±1.83 | **95.97±0.40** |
| B2 | 51.09±4.33 | 81.86±3.84 | 78.48±0.89 | 82.46±4.16 | 67.17±5.47 | **95.01±0.26** |
| B3 | 51.25±2.43 | 95.01±0.44 | 94.88±0.70 | 93.61±0.85 | 78.42±2.16 | **97.03±0.53** |
| B4 | 49.70±1.87 | 80.50±0.75 | 79.82±1.02 | 82.24±1.04 | 67.82±1.27 | **88.19±1.99** |
| B5 | 51.56±2.19 | 95.10±0.40 | 95.13±0.39 | 93.90±1.11 | 83.55±1.12 | **96.42±0.56** |
| B6 | 48.95±2.59 | 88.37±0.29 | 88.20±0.75 | 89.38±0.39 | 78.96±0.76 | **91.20±1.62** |
| S1 | 46.97±2.16 | 91.78±1.98 | 86.57±1.46 | 93.45±0.27 | 82.04±4.02 | **93.92±0.84** |
| S2 | 46.58±3.00 | 86.12±0.49 | 80.46±1.65 | 90.01±0.45 | 72.93±0.57 | **95.42±0.83** |
| S3 | 46.55±0.90 | 94.58±1.03 | 94.45±0.53 | 93.38±0.18 | 91.30±0.35 | 94.51±0.94 |
| S4 | 48.11±1.45 | 87.06±0.56 | 80.89±0.84 | 91.53±0.64 | 74.10±0.62 | **94.43±0.79** |
| S5 | 46.55±0.87 | 95.36±0.56 | 94.55±0.47 | 94.42±0.29 | 92.67±0.28 | **95.79±0.24** |
| S6 | 48.42±1.82 | 92.69±0.33 | 88.88±0.80 | 95.11±0.46 | 83.62±0.29 | **96.09±0.64** |

**(c) AUC on Fashion and LandUse21 with a missing rate of 0.3**

| | MODDIS | MODDIS | NCMOD | SRLSP | MODGD | Ours |
|---|---|---|---|---|---|---|
| F1 | 44.97±6.51 | 90.94±0.64 | 92.06±0.58 | 92.05±0.32 | 83.46±0.41 | **97.67±0.24** |
| F2 | 46.32±3.92 | 86.47±0.28 | 87.40±0.26 | 87.60±0.44 | 74.29±0.48 | **96.65±0.12** |
| F3 | 45.26±8.27 | 95.44±0.35 | 96.32±0.12 | 96.27±0.35 | 93.03±0.26 | **98.71±0.17** |
| F4 | 45.96±6.49 | 88.27±0.88 | 85.65±1.78 | 87.73±2.79 | 75.68±1.37 | **96.17±0.48** |
| F5 | 44.83±9.97 | 96.31±0.59 | 96.78±0.18 | 97.05±0.49 | 90.24±4.55 | **98.49±0.24** |
| F6 | 46.67±6.36 | 92.22±0.45 | 92.50±0.23 | 93.13±0.36 | 82.99±2.66 | **97.15±0.21** |
| L1 | 48.09±7.75 | 89.86±0.94 | 85.38±0.17 | 92.05±0.65 | 86.03±0.57 | **95.54±1.66** |
| L2 | 47.36±5.38 | 83.76±1.33 | 78.31±0.97 | 87.13±1.20 | 78.62±0.87 | **95.86±1.11** |
| L3 | 47.69±6.00 | 96.07±0.94 | 94.58±0.25 | 96.65±0.80 | 93.78±0.66 | **97.18±0.59** |
| L4 | 48.31±5.12 | 84.82±1.64 | 79.18±0.59 | 88.20±1.67 | 79.81±1.11 | **94.36±1.01** |
| L5 | 50.64±7.06 | 96.22±0.96 | 94.15±0.11 | 97.01±0.73 | 94.57±0.40 | **98.17±0.28** |
| L6 | 50.10±5.80 | 90.80±1.07 | 87.17±0.14 | 93.03±0.89 | 88.02±0.84 | **96.24±0.48** |

**(d) AUC on Fashion and LandUse21 with a missing rate of 0.45**

| | MODDIS | MODDIS | NCMOD | SRLSP | MODGD | Ours |
|---|---|---|---|---|---|---|
| F1 | 44.41±5.71 | 92.23±1.90 | 92.16±0.30 | 90.56±0.96 | 85.26±3.00 | **97.80±0.24** |
| F2 | 46.77±3.65 | 86.94±0.42 | 88.14±0.29 | 87.23±0.48 | 74.67±0.46 | **96.58±0.24** |
| F3 | 43.94±8.34 | 95.69±1.09 | 94.51±2.26 | 95.88±0.33 | 92.00±2.40 | **98.87±0.16** |
| F4 | 46.87±4.27 | 89.38±1.28 | 84.92±3.19 | 87.83±2.96 | 77.12±2.17 | **95.99±0.48** |
| F5 | 43.58±9.37 | 97.16±0.56 | 88.18±3.67 | 96.80±1.22 | 86.16±4.52 | **98.47±0.11** |
| F6 | 45.76±6.86 | 92.71±0.39 | 93.20±0.37 | 92.96±0.36 | 84.68±0.42 | **96.79±0.13** |
| L1 | 44.72±7.02 | 89.81±1.12 | 84.50±1.15 | 91.43±0.72 | 83.76±1.43 | **94.72±0.65** |
| L2 | 45.09±5.97 | 84.08±0.88 | 77.23±0.48 | 87.25±0.83 | 75.65±1.22 | **94.43±0.73** |
| L3 | 46.82±5.21 | 96.03±0.36 | 93.16±0.40 | 96.35±0.47 | 92.13±0.99 | **97.02±0.81** |
| L4 | 46.83±6.43 | 84.88±1.29 | 78.65±0.56 | 88.34±1.13 | 77.72±1.50 | **93.42±0.83** |
| L5 | 48.81±5.08 | 96.27±0.42 | 94.14±0.25 | 96.81±0.40 | 93.43±0.43 | 96.62±1.12 |
| L6 | 48.29±3.48 | 91.11±1.05 | 86.34±0.65 | 93.26±0.58 | 86.99±0.72 | **95.47±0.86** |

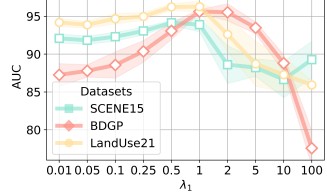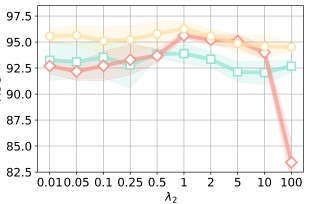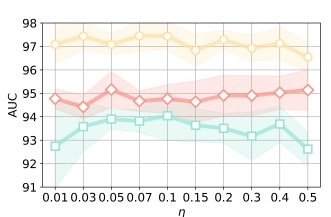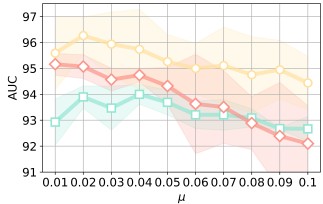

**Figure 4: Sensitivity analysis over $\lambda_1$, $\lambda_2$ $\eta$ and $\mu$ on different datasets.**

**Table 5: Ablation study on loss components.**

| | $\mathcal{L}_{oa}$ | $\mathcal{L}_{na}$ | $\mathcal{L}_{sr}$ | BDGP | SCENE15 | LandUse21 |
|---|---|---|---|---|---|---|
| (A) | | | | 21.70 | 22.59 | 36.59 |
| (B) | ✓ | | | 92.65 | 88.46 | 94.20 |
| (C) | ✓ | ✓ | | 94.84 | 90.46 | 95.37 |
| (D) | ✓ | | ✓ | 92.38 | 92.60 | 94.64 |
| (E) | | ✓ | ✓ | 86.66 | 91.43 | 93.37 |
| (F) | ✓ | ✓ | ✓ | **95.16** | **93.35** | **96.27** |

anyone of $\mathcal{L}_{oa}$, $\mathcal{L}_{na}$ and $\mathcal{L}_{sr}$ will clearly degrade the performance, indicating that all losses are indispensable in our method. On the other hand, the impact of each loss component varies across the datasets. Results of variant (B) and (E) on BDGP and LandUse21 indicate that $\mathcal{L}_{oa}$ is the most important factor in improving detection ability on these datasets, while according to variants (C) and (D),

we can find that the regularizer have a large impact on detection in SCENE15, which can also be observed in Fig. 3a .

## 5 CONCLUSION

In this paper, we propose a novel contrastive partial MVOD method named RCPMOD. Specifically, we design an outlier-aware contrastive loss with a potential outlier memory bank, ensuring that outliers are distinctly featured during the training process. A neighbor alignment contrastive loss is also proposed to learn shared local structural connections between views and this loss also enhances the effect of Cross-view Relation Transfer adopted to impute missing samples in our framework. Besides, to addresss the observed outlier overfitting phenomenon, we adopt a spreading regularization as a solution. Notably, the proposed method could also deal with outliers in the complete multi-view data. Experimental results on four benchmarks show that it can achieve the best performance under various outlier ratios and view missing rates.

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
