# OpenReview forum: "Regularized Contrastive Partial Multi-view Outlier Detection"
_acmmm.org/ACMMM/2024/Conference — MM2024 Poster_

### Official Review · Reviewer_23PW · 2024-05-21

**Rating:** 4
**Confidence:** 3

**Summary:**

This paper proposes a multi-view outlier detection method that utilizes contrastive learning to capture view-consistent information and distinguishes anomalies based on the degree of consistency. This paper also attempts to address a common issue in multi-view learning, which is outlier detection on incomplete views.

**Strengths:**

1.The authors propose a multi-view outlier detection method capable of handling three types of anomalies.

2.The authors considered the case of incomplete views and utilized cross-view relation transfer techniques to infer the missing samples.

3.The experimental results demonstrate the effectiveness of the method.

**Limitations:**

1.The authors use dual-view data as an example when introducing the method and only consider a small number of view datasets in the experiments. Additionally, since existing multi-view outlier detection datasets are artificially constructed, the authors might consider adopting an experimental setup similar to the MODGD method to test the proposed method on datasets with more views.

2.Outlier detection is a time-sensitive task with real-world significance. How is the time efficiency of the proposed method?

3.How did the authors set the missing values?

4.The authors did not provide code resources.

5.The authors should conduct an in-depth analysis of why the proposed method is superior to existing methods.

**Suitability:**

2

---

### Official Review · Reviewer_zKqv · 2024-05-24

**Rating:** 3
**Confidence:** 3

**Summary:**

This paper focuses on multi-view outlier detection, aiming to better detect class outliers and class-attribute outliers specific to multiview data. Addressing the problems of existing methods, this paper proposes a method called Regularized Contrastive Partial Multi-view Outlier Detection (RCPMOD), which thereby utilizes contrast learning to obtain view-consistent information and distinguish outliers by the consistency. Experimental results demonstrate the effectiveness of the proposed method.

**Strengths:**

Strengths
- This paper is well-organized, comprehensively analyzing the problems of existing work on multi-view outlier detection and proposing solutions.
- With clear notation and illustrative figures, the paper is easy to follow.
- The experiments on multi-view datasets seems demonstrate the effectiveness.

**Limitations:**

Limitations
- The method involves up to five losses (considering that $L_sr$ contains two terms), which makes the model very complex, and how to balance these losses poses a big challenge.
- The authors argue that partial multi-view  outlier detection is important, but there is only one piece of literature. And this issue seems to be isolated from the other contributions of the paper, which is not encouraged.
- No code or other implementation details are provided, especially considering that the proposed algorithm stacks multiple losses, which makes the experiment less convincing.

**Suitability:**

3

---

### Official Review · Reviewer_tqCd · 2024-05-25

**Rating:** 5
**Confidence:** 4

**Summary:**

In this paper, the authors study the very important problems of multi-view outlier detection, namely reducing the impact of outliers, handling varying neighborhood structures and partial multi-view data. To this end, this paper proposes a novel method named Regularized Contrastive Partial Multi-view Outlier Detection (RCPMOD). It consists of three main modules, namely Outlier-aware Contrastive Learning Module, Outlier-aware Contrastive Learning Module, and Spreading Regularization Modules. Overall, the proposed RCPMOD method is technically solid. And the authors have conducted extensive experiments to confirm the effectiveness of the proposed method. The main strengths and limitations are summarized below.

**Strengths:**

1. The problems studied in this paper are very important. In particular, reducing the impact of outliers, handling varying neighborhood structures and partial multi-view data, are important in multi-view outlier detection. And addressing these problems would significantly enhance the research progress of this field.
2. The proposed RCPMOD method is very solid. The three main modules, namely Outlier-aware Contrastive Learning Module, Outlier-aware Contrastive Learning Module, and Spreading Regularization Modules are quite interesting. And they can be integrated into the other methods.
3. The experiments are quite convincing. In particular, baselines are comprehensive, which cover different state-of-the-art methods. And the experimental datasets are sufficient. The sensitivity anlaysis and ablation study are also quite solid.

**Limitations:**

Despite the above strengths, the paper contains the following issues that must be addressed.
1. The research motivation of this work can be further enhanced. In particular, the authors are suggested to add a new subsection describing the importance of the problems to be studied in this paper. Usually, a new subsection in the beginning of section 3 is suitable. Or a new section should be generated for such purpose.
2. Before introducing the first module of the proposed method, namely "3.2 Outlier-aware Contrastive Learning", a subsection introducing the framework or the overview of this work should be provided so as to improve the readability.
3. An algorithm flowchart summarizing the main procedure of this work would be very good.

**Suitability:**

3

---

### Official Review · Reviewer_S39p · 2024-05-31

**Rating:** 4
**Confidence:** 3

**Summary:**

The paper proposes regularized contrastive partial multi-view outlier detection (RCPMOD) framework utilizes contrastive learning to learn view-consistent information and distinguish outliers, with an outlier-aware contrastive loss, neighbor alignment contrastive loss, and spreading regularization loss. RCPMOD also employs Cross-view Relation Transfer to impute missing view samples, outperforming state-of-the-art competitors on benchmark datasets.

**Strengths:**

1. The the organization of the paper is clear. Moreover, the figures are well drawn, which will illustrate the framework of the method, as well as the motivations.

2. The paper is technique soild with different aspects of experiments. Besides, the performances of the proposed model are great.

**Limitations:**

1. The main idea is not that novel, especially for constrative learning part. Can you further explain the difference between your constrastive strategy with other models?

2. I would suggest the author to open source your codes and data.

**Suitability:**

2

---

### Meta-Review · Area_Chair_SANk · 2024-07-02

**Recommendation:** Accept (Poster)
**Confidence:** 4

**Metareview:**

This paper studies  the outlier reducing issue  in multi-view outlier detection. It  learns view-consistent information via contrastive learning and distinguishs outliers using the degree of consistency. Although experiments demonstrate its effectivenss, Reviewer S39p concerns its novelty, and Reviewer tqCd suggests enhancing its motivation. Reviewer zKqv holds that the devised model is too complex such that it is hard to balance these five losses, and the experiments are less convincing. Also, there is a lack of literature on the partial multi-view outlier detection.  Reviewer 23PW states that the zero-value filling stragegy for missing values seems unconvincing. The current submitted version can be improved by adding more related works and devising more resonable filling schemes so as to enchance the convincement of this paper.

After rebutual，all of the concerns have been well addressed by the response.  So, I vote for the acceptance and encourage the authors further refine this paper into the final version.